# The Discrepancy and Agreement between Patient-Reported Percentage Pain Reduction and Calculated Percentage Pain Reduction in Chronic Pain Patients

Adam B. Fink [1,2], Charmaine Ong [1,3], Moez K. Sumar [1], Neil C. Patel [1,4] and Nebojsa Nick Knezevic [1,5,6,*]

1   Advocate Illinois Masonic Medical Center, Department of Anesthesiology, Chicago, IL 60657, USA
2   1st Faculty of Medicine, Charles University, 110 00 Prague, Czech Republic
3   Chicago Medical School, Rosalind Franklin University of Medicine and Science, North Chicago, IL 60064, USA
4   Nuvance Health Vassar Brothers Medical Center, Department of Anesthesiology, Poughkeepsie, NY 12601, USA
5   Department of Anesthesiology, University of Illinois, Chicago, IL 60612, USA
6   Department of Surgery, University of Illinois, Chicago, IL 60612, USA
*    Correspondence: nick.knezevic@gmail.com

**Abstract:** Two derivatives of the numeric rating scale (NRS) and visual analog scale (VAS), namely patient-reported percentage pain reduction (PRPPR) and calculated percentage pain reduction (CPPR), are commonly used when evaluating pain reduction. A small number of studies have attempted to assess the agreement between PRPPR and CPPR. However, they have been limited in their scope by a focus on specific types of pain, or by their focus on specific treatment modalities. As far as the authors of this article are aware, this is the first study to assess the agreement between PRPPR and CPPR in chronic pain patients, as well as the first to assess how the duration of treatment affects the correlations between PRPPR and CPPR. The aim of this retrospective analysis was to determine whether the duration of treatment affects CPPR and PRPPR, and the discrepancy and agreement between the two. Additionally, the study assessed whether individual treatment modalities, or the lack there of, impacted the discrepancy and correlation between PRPPR and CPPR. The mean PRPPR and CPPR for the entire patient population were 59.98 and 40.71, respectively. The mean discrepancy between the two parameters was 19.27. The agreement between PRPPR and CPPR, as measured by the concordance correlation coefficient, was 0.984 (95% C.I., 0.982–0.986).

**Keywords:** calculated percentage pain reduction (CPPR); patient-reported percentage pain reduction (PRPPR); patient-reported percentage improvement in pain scale (PR-PIPS); calculated percentage improvement in pain scale (C-PIPS)

## 1. Introduction

Pain is commonly assessed clinically utilizing the numerical pain rating scale (NRS) or the visual analog scale (VAS). The treatment of pain is frequently evaluated using patient-reported percentage pain reduction (PRPPR), also referred to as the patient-reported percentage improvement in pain scale (PR-PIPS), and calculated percentage pain reduction (CPPR), also known as the calculated percentage improvement in pain scale (C-PIPS). A small number of studies have attempted to assess the agreement between PRPPR and CPPR. However, they have been limited in their scope by a focus on specific types of pain, or by their focus on specific treatment modalities. As far as the authors of this article are aware, this is the first study to assess the agreement between PRPPR and CPPR in chronic pain patients, as well as the first to assess how the duration of treatment affects the correlations between PRPPR and CPPR.

Clinicians and researchers have had an interest in the clinical assessment of pain dating as far back as the 1950s [1]. Due to the subjective nature of pain, attempts to quantify it, monitor trends, and evaluate pain reduction or elevation have posed a challenge to

pain specialists and the broader medical community. The subjectivity of pain is rooted in physiology. Even though gate control theory [2] remains a widely accepted physiological model for pain conduction, the interindividual variability in the components of this system, such as the distribution of nociceptors and levels of neurotransmitters, introduces a degree of subjectivity to the perception of pain physiologically. Moreover, today it is widely accepted that sociological and psychological factors introduce further layers to the complex and subjective experience of pain [3].

A systematic review by Hjermstad et al. in 2011 concluded that the commonly used unidimensional pain scales—numerical rating scales (NRSs), verbal rating scales (VRSs), and visual analogue scales (VASs)—are all adequate for evaluating pain [1]. These findings support Williamson et al., who found all three scales to be valid, noting that each has its advantages and disadvantages. The VRS was found to be less sensitive but easier to use, while the NRS and VAS were found to be equally sensitive [4]. Statistically, the VAS was the strongest scale; however, repeat scores using the scale were found to vary up to 20% [4].

Naturally, changes in pain scales are also used to evaluate the response to the treatment of pain. In 2020, Bahreini et al. found that the minimum clinically significant difference (MCSD) required for patients in an emergency department setting to report a difference in pain was 16.55 on the VAS and 1.65 points on the NRS. This finding was irrespective of whether the patients were experiencing a reduction in or exacerbation of pain [5]. Kendrick et al. found the MSCD to be 1.39 on the NRS scale in the emergency department setting regardless of the etiology of pain [6]. When evaluating patients suffering from unspecified neck pain, Kovacs et al. found that the improvements on the NRS of less than 1.5 to be irrelevant [7].

Additionally, a derivative of pain scales is often used when evaluating pain reduction, known as calculated percentage pain reduction (CPPR). CPPR is calculated by subtracting the post-treatment NRS score from the pretreatment NRS score and dividing the result by the pretreatment score—(pre-NRS—post-NRS)/pre-NRS [4]. Another commonly used metric is patient reported pain percentage reduction (PRPPR). PRPPR is generated by asking patients to rate their perceived improvement of pain as a percentage, scored from 0–100%, or via a similarly phrased question. The percentage of pain reduction considered to be significant has been placed at 50% [8], 33% [9], and 30% [10] in different studies.

Intuitively, it would be expected that PRPPR and CPPR would reflect very similar—if not identical—results; however, the limited number of studies have demonstrated that there is in fact a statistically significant discrepancy between them as well as an imperfect agreement between the two [11–14].

## 2. Materials and Methods

Following approval from the Advocate Healthcare Institutional Review Board on 13 November 2018 as protocol number 6985, the authors of this article conducted a retrospective analysis of 1362 patients treated for various chronic pain conditions. In-depth chart reviews of patients at the Chicago Anesthesia Pain Specialists Clinic were utilized to collect demographics, treatments, and treatment responses. All patients were treated for at least 6 months and were seen in the clinic no less than 4 times. Patient health information was protected via utilization of secure computers. Race/ethnicity was recorded and categorized into White non-Hispanic, White-Hispanic, African American, Asian, or other. The demographic data of the patients' studies are summarized in Table 1. Patients were interviewed at each visit and were asked to quantify pain on a numeric rating scale (NRS) for both pre- and post-treatment pain scores, as well as subjective percentage improvement. Various pharmaceutical and interventional treatment modalities were identified and recorded. Opioid utilization at any time throughout treatment was identified, and morphine milligram equivalents (MMEs) at first and last visit were calculated using MDCalc (11). Patients were stratified into six groups determined by the duration of their treatment. The IBM SPSS 27 software (IBM Corporation, Armonk, NY, USA) was used to analyze the collected data. Frequency tables were utilized to identify proportions of patients with various nominal

variables. One-sample *t*-tests were utilized to calculate mean numerical variables, and the paired *t*-test was used to assess the difference between PRPPR and CPPR for the entire patient population and the stratified groups. The concordance correlation coefficient was used to assess the agreement between PRPPR and CPPR.

**Table 1.** Patient demographic data.

| Demographic | | Patient (*n* = 1362) |
|---|---|---|
| Age (mean) | | 63.01 |
| | >90 | 2.9% |
| | 80–89 | 9.9% |
| | 70–79 | 19.7% |
| | 60–69 | 27% |
| | 50–59 | 23.4% |
| | 40–49 | 10.8% |
| | <39 | 6.1% |
| Sex | | |
| | Male | 39.2% |
| | Female | 60.8% |
| Intervention | | |
| | Opioid pain medications | 65.2% |
| | Steroid injections | 88.4% |
| | Gabapentin | 39.5% |
| | NSAIDs | 53.7% |
| | Muscle relaxants | 30.5% |
| | Tricyclic antidepressants | 6.2% |
| | Benzodiazepines | 22.7% |
| | Other psychiatric medications | 26.9% |
| | Other interventional therapy | 26.6% |

To assess how different treatments may impact the discrepancy and agreement between the PRPPR and CPPR, patients were categorized using medications or other interventions as categorical variables, namely, opioid pain medications, steroid injections, gabapentin, NSAIDs, muscle relaxants, tricyclic antidepressants, benzodiazepines, other psychiatric medications, and other interventions. Once stratified into groups, the paired-sample *t*-test was used, generating the mean PRPPR, mean CPPR, and the correlation between the two parameters.

## 3. Results

The mean PRPPR and CPPR for the entire patient population were 59.98 and 40.71, respectively. The mean discrepancy between the two parameters was 19.27. The agreement between PRPPR and CPPR, as measured by the concordance correlation coefficient, was 0.984 (95% C.I., 0.982–0.986).

When stratified into groups based on the duration of treatment, the PRPPR and CPPR were 56.7 and 37.94, respectively, amongst patients treated for less than 1 year. The mean discrepancy amongst this cohort was 18.75, while the correlation coefficient was 0.989 (95% C.I., 0.978–0.995).

The PRPPR and CPPR were 61.45 and 42.18, respectively, among patients treated for 1–2 years. The average difference among this group was 19.26, and the concordance correlation efficient was 0.989 (95% C.I., 0.987–0.991).

The PRPPR and CPPR for patients treated for 2–3 years were 54.54 and 35.94, respectively. The mean discrepancy for this cohort was 18.60, and the concordance correlation coefficient was found to be 0.976 (95% C.I., 0.970–0.980).

Among patients treated for 3–4 years, the PRPPR and CPPR were 48.87 and 30.37, respectively. The mean discrepancy within this group was 18.49 and the concordance correlation coefficient was 0.979 (95% C.I., 0.973–0.983).

The PRPPR and CPPR were recorded as 66.86 and 46.81, respectively, among patients treated for 4–5 years. The average difference among this group was 20.02, and the concordance correlation coefficient was calculated as 0.983 (95% C.I., 0.974–0.989).

For patients treated for longer than 5 years, the PRPPR and CPPR were found to be 81.66 and 60.14, respectively. The mean discrepancy within this cohort was 21.51, while the concordance correlation coefficient was determined to be 0.980 (95% C.I., 0.974–0.985). The fluctuations in PRPPR and CPPR over time are depicted graphically in Figure 1, and the regression curve between the two is depicted in Figure 2.

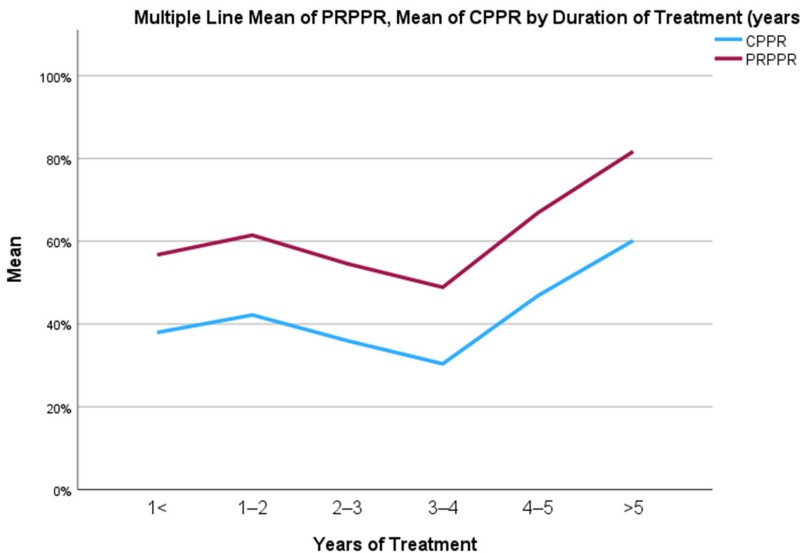

**Figure 1.** The fluctuations in PRPPR and CPPR over time.

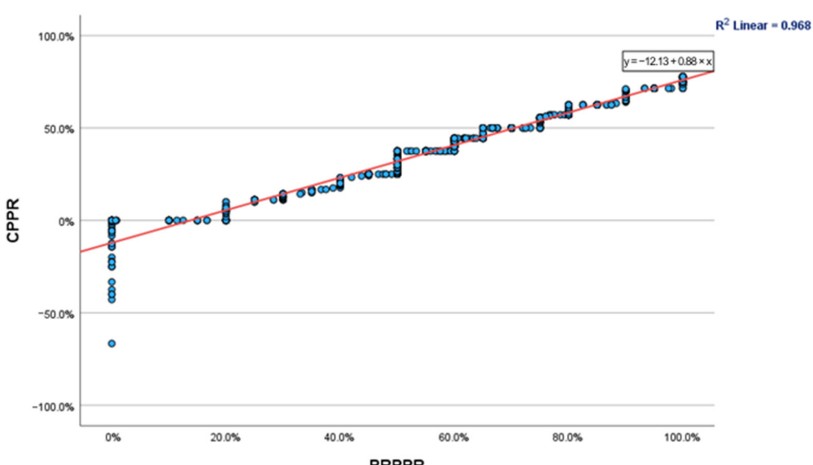

**Figure 2.** Scatter plot and linear regression curve of CPPR and PRPPR.

When the patient population was restratified into groups based on the presence or absence of certain medications or treatment modalities in their therapeutic regimen, the following results were found. Patients utilizing opioid pain medications reported a PRPPR of 55.12 and a CPPR of 36.38, accounting for a mean discrepancy of 18.73. The correlation between PRPPR and CPPR was 0.982 ($p < 0.001$). Patients who did not receive opioid pain medication reported a mean PRPPR and mean CPPR of 68.96 and 48.71, respectively, providing for a discrepancy of 20.25, with a correlation of 0.985 ($p < 0.001$).

Individuals who received Steroid injections had a PRPPR of 60.26 and a CPPR of 40.99, with an average difference of 19.26. The correlation between PRPPR and CPPR was 0.984 ($p < 0.001$). Patients who did not receive steroid injections had a mean PRPPR and mean CPPR of 57.45 and 38.22 respectively, resulting in a difference of 19.23, with a correlation of 0.984 ($p < 0.001$).

When using gabapentin, patients experienced a PRPPR of 59.11 and a CPPR of 39.95, producing a mean difference of 19.15. The correlation between the two measures was 0.981

($p < 0.001$). Patients who did not use gabapentin had a mean PRPPR and mean CPPR of 60.62 and 31.30, respectively, leading to a difference of 19.32, with a correlation of 0.986 ($p < 0.001$).

The PRPPR and CPPR were reported to be 62.32 and 42.73, respectively, by patients using NSAIDs, with a mean discrepancy of 19.58. The correlation between PRPPR and CPPR was 0.985 ($p < 0.001$). Patients not using NSAIDs had a mean PRPPR and mean CPPR of 57.36 and 38.49, respectively, resulting in a difference of 18.86, with a correlation of 0.984 ($p < 0.001$).

Patients who received muscle relaxants had a mean PRPPR of 58.25 and a mean CPPR of 39.33, and thus a difference of 18.95. The correlation between the two measures was 0.987 ($p < 0.001$). Conversely, patients who did not take Muscle relaxants had a mean PRPPR and mean CPPR of 60.76 and 41.37, resulting in a difference of 19.38, with a correlation of 0.983 ($p < 0.001$).

Amongst patients utilizing TCAs, a mean PRPPR of 66.12 and a mean CPPR of 46.01 were reported, generating a discrepancy of 20.10 with a correlation of 0.984 ($p < 0.001$). Patients who did not utilize TCAs reported a PRPPR of 59.62 and a CPPR of 40.43, resulting in a discrepancy of 19.19 with a correlation of 0.984 as well ($p < 0.001$).

Benzodiazepine-receiving patients reported a PRPPR and CPPR of 55.02 and 36.48, respectively. The discrepancy was therefore found to be 18.85, while the correlation was found to be 0.987 ($p < 0.001$). Patients who did not receive benzodiazepines were found to have a PRPPR of 61.49, a CPPR of 42.03, a discrepancy of 19.46, and a correlation of 0.984 ($p < 0.001$).

A PRPPR of 59.86 and CPPR of 40.57 were reported in patients using other psychiatric mediations. The discrepancy between the two parameters in this population was thus 19.28, and the correlation was found to be 0.987 ($p < 0.001$). Patients not receiving other psychiatric medications had a mean PRPPR and mean CPPR of 60.09 and 40,84, respectively, resulting in a difference of 19.24, with a correlation of 0.984 ($p < 0.001$).

The subset of patients who were treated with other interventions reported a discrepancy between PRPPR and CPPR of 19.06, caused by a PRPPR of 60.96 and a CPPR of 41.90. The correlation between the parameters was 0.989 ($p < 0.001$). Patients who did not receive other forms of intervention had a mean PRPPR and CPPR of 59.56 and 40.23, respectively, with a difference of 19.33 and correlation of 0.982 ($p < 0.001$).

A summary of the effect of different treatment modalities on PRPPR, CPPR, and the correlation between the two metrics can be found in Table 2.

**Table 2.** The effect of medications on the PRPPR, CPPR, discrepancy, and Correlation.

| Intervention | PRPPR | CPPR | Discrepancy | Correlation |
|---|---|---|---|---|
| Opioid pain medication | 55.12 | 36.38 | 18.73 | 0.982 ($p < 0.001$) |
| No opioid pain medication | 68.96 | 48.71 | 20.25 | 0.985 ($p < 0.001$) |
| Steroid injections | 60.26 | 40.99 | 19.26 | 0.984 ($p < 0.001$) |
| No steroid injections | 57.45 | 38.22 | 19.23 | 0.984 ($p < 0.001$) |
| Gabapentin | 59.11 | 39.95 | 19.15 | 0.981 ($p < 0.001$) |
| No gabapentin | 60.62 | 41.30 | 19.32 | 0.986 ($p < 0.001$) |
| NSAID | 62.32 | 42.73 | 19.58 | 0.985 ($p < 0.001$) |
| No NSAID | 57.36 | 38.49 | 18.86 | 0.984 ($p < 0.001$) |
| Muscle relaxants | 58.28 | 39.33 | 18.95 | 0.987 ($p < 0.001$) |
| No muscle relaxants | 60.76 | 41.37 | 19.38 | 0.983 ($p < 0.001$) |
| Tricyclic antidepressants | 66.12 | 46.01 | 20.10 | 0.984 ($p < 0.001$) |
| No tricyclic antidepressants | 59.62 | 40.43 | 19.19 | 0.984 ($p < 0.001$) |
| Benzodiazepines | 55.02 | 36.48 | 18.54 | 0.987 ($p < 0.001$) |
| No benzodiazepines | 61.49 | 42.03 | 19.46 | 0.984 ($p < 0.001$) |
| Other psychiatric medications | 59.86 | 40.57 | 19.28 | 0.987 ($p < 0.001$) |
| No other psychiatric medications | 60.09 | 40.84 | 19.24 | 0.984 ($p < 0.001$) |
| Other interventions | 60.96 | 41.90 | 19.06 | 0.989 ($p < 0.001$) |
| No other interventions | 59.56 | 40.23 | 19.33 | 0.982 ($p < 0.001$) |

## 4. Discussion

As far as the authors of this paper are aware, this study is the first to evaluate PRPPR, CPPR, and the agreement between the two in chronic pain patients treated over the course of several years. Several noteworthy publications have described similar studies conducted in smaller patient populations who suffered from specific forms of acute pain or were suffering from chronic pain but studied for a short amount of time. Hagedorn et al. published a study of 174 patients in 2021 who were treated with spinal cord stimulator implants for chronic pain between 2017–2019 and concluded that although the PR-PIPS and C-PIPS were highly correlated, a substantial disagreement existed between the two methods after the concordance correlation coefficient was calculated at 0.76 (95% C.I., 0.69 to 0.81). The mean C-PIPS was 54 with a standard deviation of 28, while the mean PR-PIPS was 59 with a standard deviation of 25 [11].

In 2017, Pratici et al. published an article detailing the differences in CPPR and PRPPR obtained from 97 patients who received epidural analgesia during labor and were asked to rate their pain just prior to the administration on analgesia and 30 min thereafter. The study reported a PRPPR of 79 with a standard deviation of 21.5 and a CPPR of 80 with a standard deviation of 21.2. The concordance correlation coefficient was calculated using both the VAR and NRS scores and yielded a coefficient of 0.76 (95% CI 0.6 to 0.8) and 0.77 (95% C.I., 0.6 to 0.8), respectively. The authors concluded that moderate agreement existed between PRPPR and CPPR [12].

The discrepancy between PRPPR and CPPR amongst 197 patients receiving first time fluoroscopic steroid injections for musculoskeletal or radicular pain was described by Cushman et al. in 2015. The patients in this study were stratified in several ways: patients with reported pain improvement vs. patients who reported no improvement of pain; male vs. female; and by age groups (18–40, 41–60, 61+). The authors reported several noteworthy findings. First, there was a fair-to-moderate association between PRPPR and CPPR at 3 weeks follow-up. Second, patients reported a higher PRPPR in comparison to their CPPR more than twice as often, leading the authors to conclude that patients are unable to assess their PRPPR compared to their preintervention pain score with a high degree of accuracy. The mean difference between PRPP and CPPR was +16% (95% C.I., +11% to 21%). The concordance correlation coefficient was 0.44 (95% C.I., 0.35–0.54) [13].

Seven-hundred and sixty-one patients with acute pain or cancer pain were included in a study published in 2003 by Cepeda et al. Patients' CPPR and PRPPR were evaluated in the acute setting, while receiving analgesics every 10 min until their pain intensity fell below 4/10 using a verbal NRS scale. This study found the concordance correlation coefficient to be 0.56 (95% C.I., 0.54–0.58), with the authors concluding that a good [15] agreement existed between PRPPR and CPPR in this study. The mean difference between PRPPR and CPPR was −2.6 with the 95% limits of agreement for the difference between these two measures at −12 to 17%. Additionally, CPPR was found to underestimate PRPPR at higher levels; however, this did not translate into a clinically significant difference [14].

The current study found that PRPPR overestimates CPPR in keeping with the above publications, thus validating this phenomenon. However, in contrast to previous studies, the current study found much higher levels of agreement between PRPPR and CPPR. The levels of agreement previously described were moderate to strong, while this study found very strong levels [15] of agreement between the two, despite a larger overall discrepancy. Based on these findings, two outstanding questions require further investigation. First, why does PRPPR overestimate CPPR? Second, which of the measurements is a more accurate representation of pain reduction?

Regarding the first question, it is worth noting that when calculating [4] CPPR at any point in time, patients are required to only report the intensity of pain they are currently experiencing, as the pretreatment score was recorded prior to treatment. However, when patients are asked to produce a PRPPR, they are required not only to assess their current intensity of pain, but also to recall their pain levels prior to treatment and deduce

their improvement as a percentage. Naturally this renders PRPPR more susceptible to subjectivity.

In 1981, Linton et al. published a study that had recruited 12 chronic pain patients to evaluate the accuracy of memory for chronic pain. The mean duration of pain experienced by the patients was 2 years, with a minimum duration of 6 months. Ninety-two percent (11/12) of the patients recalled higher levels of pain than those originally recorded as a baseline, reporting levels 19% higher [16]. This phenomenon serves as a plausible explanation for the overestimation of CPPR by PRPPR, which has now been reproduced in our study as well. Hypothetically, if a patient's baseline NRS score is 7, and at the conclusion of their treatment they report it as 3, their CPPR would be 57%. However, when asked to produce a PRPPR, if they overestimate their baseline NRS score by 19%, they should recall their baseline as a score of 8.3. If they then use the same end-of-treatment NRS score of 3, the PRPPR should produce a value of roughly 64%. This, of course, assumes that patients are using a mathematical formula for CPPR in their mental processing when asked to produce a PRPPR, which is unlikely. However, this hypothetical exercise does provide a mathematical and logical explanation to the phenomenon now validated by multiple studies.

Several later studies highlighted the inaccuracy of patient recollection of pain and identified factors that may play a role in distortion of pain recollection. Eich et al. found that current levels of pain intensity alter the recollection of pain. Patients who experience higher intensities of pain at the present time overestimate their baseline pain, and patients who experience lower pain intensities currently underestimate their initial pain [17]. In addition, Jamison et al. described how various types of pain, as well as psychological and social factors, reduce the accuracy of pain recollection. Perhaps most interestingly, patients who relied on tranquilizers and sleeping medication to manage their pain were amongst the most likely to overestimate previously recorded levels [18]. Easton et al. found that even in the acute setting, post-trauma patients' recollection of pain is unreliable [19]. Furthermore, there is evidence to suggest that the recall of chronic pain is less reliable than that of acute pain [20].

After reviewing the body of work presented in this article, the authors suggest that the underlying cause of the discrepancy between PRPPR and CPPR is inaccurate patient recollection of pain, likely overestimating past levels. However, it remains unclear which metric, PRPPR or CPPR, is a better tool for evaluating pain reduction in chronic pain patients, and this requires further research.

This predicament is best illustrated when observing the phenomena clearly illustrated in Figure 2. Despite a very strong correlation coefficient for the entire patient population, patients who reported a PRPPR of approx. 50% had CPPRs ranging between 30–45%, while patients whose CPPR was approx. 40% reported PRPPRs of 45–65%. Given that previous studies [8–10] have concluded that the percentage of pain improvement or reduction required to substantiate a clinically significant difference may be as low as 30% or as high as 50%, the use of PRPPR or CPPR in isolation is likely to miss a subset of patients who may not have experienced a clinically significant improvement.

Regarding the effect that various medications and interventions may have on the discrepancy between PRPPR and CPPR, and the correlation between the two, this study found no significant impact.

## 5. Conclusions

PRPPR overestimated CPPR in the patient population of this study. The mean discrepancy between the two fluctuated minimally when patients were stratified into groups based on the duration of their treatment despite large fluctuations in both PRPPR and CPPR. Furthermore, the agreement between the patients' PRPPR and CPPR was very strong and fluctuated minimally with the duration of treatment. The lowest value for the concordance correlation coefficient was 0.976, found in the group of patients treated for 2–3 years.

As the agreement between PRPPR and CPPR is very strong, it can be inferred that any increase or decrease in either of the parameters may be predicted in the other with a high degree of accuracy. However, it is not possible to deduce which of the parameters is a more accurate representation of actual improvement in a patient's underling pain, and further research is required to gain insight into the mechanisms driving the discrepancy between PRPPR and CPPR.

**Author Contributions:** Conceptualization, A.B.F. and N.N.K.; methodology, A.B.F., C.O., M.K.S., N.C.P. and N.K; validation, A.B.F., C.O., M.K.S. and N.N.K.; formal analysis, A.B.F. and N.N.K.; investigation, A.B.F., C.O., M.K.S. and N.N.K.; resources, A.B.F., C.O., M.K.S., N.C.P. and N.N.K.; data curation, A.B.F., N.C.P. and N.N.K.; writing—original draft preparation, A.B.F. and N.N.K.; writing—review and editing, A.B.F. and N.N.K.; visualization, A.B.F., C.O., M.K.S. and N.N.K.; supervision, N.N.K. All authors have read and agreed to the published version of the manuscript.

**Funding:** This research received no external funding.

**Institutional Review Board Statement:** The study was conducted in accordance with the Declaration of Helsinki and approved by the Institutional Review Board of Advocate Healthcare as protocol number 6985 on 13 November 2018.

**Informed Consent Statement:** This study was approved for waiver of consent and HIPAA waiver of authorization under expedited criterion #5, research involving data collected for any purpose, not only for research purposes.

**Data Availability Statement:** Data are contained within the article.

**Acknowledgments:** The authors wish to acknowledge and thank Noam Shimoni for assisting with the statistical analysis.

**Conflicts of Interest:** The authors declare no conflict of interest.

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
