# Peer review of "The Discrepancy and Agreement between Patient-Reported Percentage Pain Reduction and Calculated Percentage Pain Reduction in Chronic Pain Patients"

_2035-8377, doi:10.3390/neurolint15020034_

Round 1
Reviewer 1 Report
In this manuscript, Fink et al., demonstrates the discrepancy in the PRPPR and CPPR in chronic pain patients. This agrees with previous publications in the same field and authors confirming the previous reports with higher sample size. This is an interesting manuscript, which is overall very well written and documented.
Have you noticed any relationship between both PRPPR and CPPPR and the age of the patients. If so, please add a graphical data for the same. Authors must mention more details about the age of patients. Are all 1362 patients having same age.
Author Response
We sincerely thank you all for your feedback.
Patient age and influence on PRPPR and CPPR – the mean age of the entire patient population is 63.01 and therefor appeared as 63 in the original manuscript. That has now been corrected to 63.01. As for the influence of age and its effect on our study, we did not test for this. However, we have added a breakdown of the patient age demographics to the demographic table based on you feedback.
Reviewer 2 Report
The text is well written and the reader enjoys reading it. The input is adequate and developed as intended.
I would just like to share a few thoughts.
Why was it necessary to know the ethnicity-race of the population you examined? What role did it play in the results of the study?
The same applies to BMI
Patients were divided into groups according to the duration of treatment and not the type of drugs used. Noting that the table used psychiatric medications as well as opioids, do you think these patients are reliably expressing their pain intensity?
Given that both patients were interviewed at each visit and asked to quantify pain on a numeric rating scale (NRS) for both pre- and post-treatment pain scores and subjective percent improvement, should they be excluded?
Author Response
We sincerely thank you all for your feedback.
Data regarding patient BMI and Ethnicity was collected as part of the general data gathered for the study although we did not use it in the context of our study design. As such, we agree with your feedback and have omitted these parameters from the demographic data table.
Your point regarding the reliability of patient reported pain intensity in the subset of patients utilizing opioids and psychiatric medications is pertinent. Please see the additional information added to the results section as patients on these medications reported a PRPPR and CPPR with a discrepancy and correlation highly similar to that of patients not utilizing these medications.
Based on the results, the conclusion we feel is better substantiated, is that the main driver behind the phenomenon of the discrepancy between PRPPR and CPPR is erroneous patient recall of pain intensity as opposed to erroneous reporting of current levels.